# Metagenomic Insights into Disease-Induced Microbial Dysbiosis and Elemental Cycling Alterations in *Morchella* Cultivation Soils: Evidence from Two Distinct Regions

**DOI:** 10.3390/jof11090663

**Published:** 2025-09-10

**Authors:** Zong-Lin Deng, Feng-Ming Yu, Xiang Ma, Qi Zhao, Jian-Kui Liu

**Affiliations:** 1The Clinical Hospital of Chengdu Brain Science Institute, School of Life Science and Technology, University of Electronic Science and Technology of China, Chengdu 611731, China; dengzonglin111@163.com (Z.-L.D.); fungixiangm@163.com (X.M.); 2Key Laboratory of Phytochemistry and Natural Medicines, Chinese Academy of Sciences, Kunming 650201, China; yufengming@mail.kib.ac.cn

**Keywords:** *Morchella* cultivation, soil microbiome, microbial dysbiosis, metagenomics, rhizosphere microbiome, co-occurrence network, nutrient metabolism

## Abstract

Soil-borne diseases represent a major constraint on the sustainable cultivation of morel mushrooms (*Morchella* spp.), yet the microbial ecological mechanisms driving disease occurrence and progression remain poorly understood. In this study, we conducted comparative metagenomic analyses of rhizosphere and root-adhering soils associated with healthy and diseased *Morchella* crops from two major production regions in China, aiming to elucidate shifts in microbial community composition, assembly processes, and functional potential. Disease conditions were linked to pronounced microbial dysbiosis, with community assembly shifting from stochastic to deterministic processes, particularly within fungal communities under host selection and pathogen pressure. Co-occurrence network analysis revealed substantial reductions in connectivity, modularity, and clustering coefficients in diseased soils, indicating the loss of ecological stability and keystone taxa. Functional annotations using CAZy, COG, and KEGG databases showed that healthy soils were enriched in genes related to carbohydrate metabolism, aerobic respiration, and ecosystem resilience, whereas diseased soils exhibited higher abundance of genes associated with stress responses, proliferation, and host defense. Furthermore, elemental cycling analysis demonstrated that healthy soils supported pathways involved in aerobic carbon degradation, nitrogen fixation, phosphate transport, and sulfur oxidation, while diseased soils favored fermentation, denitrification, phosphorus limitation responses, and reductive sulfur metabolism. Collectively, these results highlight the importance of microbial functional integrity in maintaining soil health and provide critical insights into microbiome-mediated disease dynamics, offering a foundation for developing microbiome-informed strategies for sustainable fungal crop management.

## 1. Introduction

Morel mushrooms (*Morchella* spp.), prized for their distinctive flavor and rich bioactive compounds—including polyphenols and terpenoids with antioxidant and immunomodulatory properties—are a high-value commodity in the global edible mushroom market [1,2,3]. Recent advances in artificial cultivation have spurred rapid expansion of *Morchella* farming, particularly in southwestern China, where provinces like Sichuan, Yunnan, and Guizhou are key production hubs [4,5]. However, this rapid growth faces persistent challenges, notably unstable yields, high rates of malformed fruiting bodies, and recurrent outbreaks of devastating soil-borne diseases. Pathogens such as *Fusarium*, *Paecilomyces*, *Diploöspora*, and *Cladobotryum* have been consistently implicated in severe fruiting body decay, significantly constraining both productivity and economic sustainability [6,7,8,9,10,11].

Mounting evidence suggests that beyond direct pathogenic effects, these disease outbreaks are closely tied to disruptions in the soil microbiome. The rhizosphere and root-adhering soil represent critical microhabitats where host–microbe–soil interactions occur, directly influencing nutrient exchange, pathogen suppression, and fungal development [12,13,14,15,16]. While prior research has demonstrated that microbial communities affect mycelial growth, nutrient acquisition, and fruiting body formation via modulation of rhizosphere chemical signaling and nutrient fluxes [17,18,19], most studies have predominantly focused on substrate optimization, environmental controls (e.g., moisture, temperature), and pathogen identification [[8],[9],[11],[16],[20],[21],[22],,[23]]. The ecological mechanisms underpinning how microhabitat-specific microbial dynamics influence morel health and disease resistance remain poorly elucidated.

With the advent of metagenomic sequencing and high-resolution ecological modeling, microbial research has progressed beyond taxonomic surveys to functional and metabolic network analyses [24,25,26,27,28,29,30,31,32]. Integrated approaches combining metagenomics, functional annotation, and co-occurrence network analysis have proven effective in revealing how microbial communities mediate biogeochemical cycling of key elements—including carbon (C), nitrogen (N), phosphorus (P), and sulfur (S)—while maintaining ecosystem resilience [33,34,35,36]. Despite these advancements, a critical knowledge gap persists regarding how soil microbiomes in morel cultivation systems respond to health deterioration and pathogen invasion, particularly in terms of functional reprogramming, community stability, and nutrient cycling disruption.

To address these gaps, we conducted a comprehensive metagenomic investigation in two representative *Morchella* cultivation regions in China: Yongsheng (Lijiang, Yunnan; plateau monsoon climate, 2400 m elevation) and Qianxi (Bijie, Guizhou; karst terrain, ~1200 mm annual rainfall). These two sites exhibit distinct soil physicochemical properties and climatic regimes, providing an ideal natural framework for comparative ecological analysis. A systematic sampling design was employed, targeting five soil compartments: healthy rhizosphere (H), healthy root-adhering soil (H_R), diseased rhizosphere (D), diseased root-adhering soil (D_R), and unplanted control soils (CK). Leveraging metagenomic sequencing and integrative ecological analyses, this study aimed to: **(i)** characterize changes in microbial taxonomic diversity and community composition in response to host health status and microhabitat variation; **(ii)** identify region-specific microbial biomarkers and core functional modules that distinguish healthy from diseased soils, and elucidate the ecological processes—stochastic versus deterministic—that govern bacterial and fungal community assembly; and **(iii)** investigate how disease-induced microbial dysbiosis disrupts key nutrient cycling pathways, with a focus on carbon, nitrogen, phosphorus, and sulfur metabolism. Our findings provide novel insights into the microbial ecological mechanisms underlying soil health in *Morchella* cultivation and establish a foundation for microbiome-informed disease monitoring and sustainable management strategies.

## 2. Materials and Methods

### 2.1. Soil Sampling and Experimental Design

Soil samples were collected from two major *Morchella* cultivation regions in China: Yongsheng (YS), Yunnan Province (plateau monsoon climate; 2400 m elevation) and Qianxi (QX), Guizhou Province (karst terrain; ~1200 mm annual rainfall). To capture microbial variations associated with cultivation practices and host health status, five soil microhabitats were defined: (i) CK: control soil from adjacent uncultivated plots with no prior *Morchella* cultivation; (ii) H: rhizosphere soil surrounding healthy fruiting bodies; (iii) D: rhizosphere soil from diseased fruiting bodies exhibiting visible symptoms; (iv) H_R: root-adhering soil from healthy fruiting bodies obtained by gentle brushing; and (v) D_R: root-adhering soil from diseased fruiting bodies collected using the same method (Figure 1a). Each microhabitat type was sampled in triplicate per region, resulting in a total of 30 samples. These triplicates represent true biological replicates, each collected from independent cultivation beds within the same farm region, separated by at least 5–10 m to minimize microheterogeneity. All samples were aseptically collected during the fruiting stage using sterile tools to prevent cross-contamination. Soil was collected from a depth of 0–10 cm to ensure representativeness. Immediately after collection, samples were flash-frozen in liquid nitrogen and stored at −80 °C until further processing to preserve microbial community structure and function.

### 2.2. DNA Extraction and Metagenomic Sequencing

Total genomic DNA was extracted from 0.5 g of soil using the TGuide S96 Magnetic Soil/Stool DNA Kit (TIANGEN Biotech, Beijing, China) following the manufacturer’s protocols. The concentration and purity of the extracted DNA were quantified using a Qubit 4.0 Fluorometer (Thermo Fisher Scientific, Waltham, MA, USA), and its integrity was assessed by 1% agarose gel electrophoresis. Metagenomic libraries were constructed using the NEBNext^®^ Ultra™ DNA Library Prep Kit for Illumina (New England Biolabs, Ipswich, MA, USA). Paired-end sequencing (2 × 150 bp) was conducted on the Illumina NovaSeq 6000 platform (BGI-Wuhan, China) in PE150 mode, generating an average of approximately 33.5 million raw reads per sample (~5.0 Gb/sample). The sequencing depth was chosen to ensure sufficient coverage for accurate microbial community profiling and functional annotation. Quality metrics of the sequencing data confirmed high reliability across all samples, with GC content ranging from 59.38% to 63.5%, Q20 values exceeding 96.5%, and Q30 values above 90% (Appendix A).

### 2.3. Quality Control and Metagenomic Assembly

Raw reads underwent stringent processing with fastp v0.23.2, including adapter removal, quality trimming (sliding window: 4 bp, mean quality < Q20), and elimination of reads < 50 bp. Host-derived sequences were removed by alignment to the NCBI NT database using BWA v0.7.17, and rRNA was filtered with SortMeRNA v4.3. After filtering, clean data ranged from 3.5 to 5.0 Gb per sample. De novo assembly was conducted with MEGAHIT v1.2.9 (k-mer step: 21), and contigs shorter than 500 bp were discarded. The assembly quality (mean N50: 4267 bp; maximum contig length > 516 kb) supported reliable downstream analyses.

### 2.4. Gene Prediction and Functional Annotation

Open reading frames (ORFs) were predicted from assembled contigs using MetaGeneMark v3.38. The resulting ORFs were clustered with CD-HIT v4.8.1 at 95% sequence identity and 90% alignment coverage to generate a non-redundant gene catalog (Appendix A). Functional annotation was performed against the KEGG (version 90.3), eggNOG (version 5.0), CAZy (version 2022), and NR (version 2022-06) databases using DIAMOND v2.0.15 (e-value ≤ 1 × 10^−5^). The specific database versions were selected to ensure accuracy and reproducibility of functional assignments.

### 2.5. Taxonomic Profiling and Abundance Estimation

Taxonomic annotation was performed by aligning non-redundant gene catalog sequences against the NCBI NR database with taxonomic subsets including bacteria, fungi, archaea, and viruses. The alignment was executed using DIAMOND (blastp, e-value ≤ 1 × 10^−5^). For each gene, hits with e-value ≤ (minimum e-value × 10) were retained for subsequent taxonomic classification. To resolve multiple annotations, the Lowest Common Ancestor (LCA) algorithm, implemented in MEGAN v6.18.2, was used to assign the most precise taxonomic rank before branching divergence. Based on the LCA results combined with gene abundance tables, taxonomic abundance profiles were generated at six hierarchical levels: kingdom, phylum, class, order, family, genus, and species. The abundance of each taxon in a given sample was calculated as the cumulative abundance of genes annotated to that taxon. In parallel, the number of genes assigned to each taxonomic unit was also computed; this was defined as the count of non-zero abundance genes within that taxon for each sample.

### 2.6. Diversity Analysis

Alpha diversity indices (Chao1, Shannon, Simpson) were calculated via QIIME2. Beta diversity assessment employed NMDS ordination based on Bray–Curtis dissimilarities, with group differences tested by PERMANOVA.

### 2.7. Differential Abundance and Functional Pathway Analysis

Taxonomic biomarkers were identified by LEfSe (LDA score > 2.5, *p* < 0.05). Functional pathway differences were evaluated using DESeq2 and Welch’s t-test with Benjamini–Hochberg FDR correction. Carbon, nitrogen, phosphorus, and sulfur cycling pathways were extracted from KEGG annotations and visualized in STAMP.

### 2.8. Co-Occurrence Network Analysis

Genus-level co-occurrence networks were constructed based on significant Spearman correlations (|*r*| > 0.5, FDR < 0.01) to identify potential microbial interactions. The Spearman correlation coefficient threshold of 0.5 was chosen to capture strong co-occurrence relationships, and the FDR threshold of 0.01 was applied to control for multiple testing and minimize false positives. Network visualization and topological analysis (e.g., degree centrality) used Gephi v0.10.1 with Fruchterman Reingold layout and modularity-based community detection to identify keystone taxa.

### 2.9. Statistical Analysis and Visualization

All statistical analyses and visualizations were performed in R v4.2.1. Data visualization utilized ggplot2 and pheatmap packages, while ecological statistics relied on vegan. Group comparisons applied Student’s *t*-test or Wilcoxon rank-sum test based on data distribution and variance homogeneity.

## 3. Results

### 3.1. Experimental Design and Metagenomic Data Summary

Pairwise Spearman correlation analysis of non-redundant gene abundances indicated that the microbial community structure was significantly influenced by both geographic location (*p* < 0.001, PERMANOVA) and soil microhabitat (*p* = 0.003). Samples formed distinct clusters based on region and microhabitat in hierarchical clustering analysis (Figure 1a,b). Community composition analysis at the kingdom level further revealed that bacteria dominated across all samples (relative abundance > 89%), with minor contributions from Archaea (<6%), Eukaryota (<4%), and Viruses (<1%) (Figure 1c).

Metagenomic assembly revealed 6001 conserved core genes across all samples, alongside treatment-specific gene subsets (Figure 1d), suggesting a stable functional backbone with context-responsive genetic modules.

Based on the metagenomic data, eight medium-to-high-quality microbial genome bins (MAGs) were reconstructed (completeness >70%, contamination <5%). Phylogenetic analysis indicated that these MAGs belonged to the phyla *Proteobacteria* (*n* = 5), *Actinobacteriota* (*n* = 2), and *Patescibacteria* (*n* = 1) (Figure 1e; Appendix A). These MAGs represent phylogenetically diverse bacterial lineages and provide a valuable resource for exploring microbial functions potentially linked to *Morchella* cultivation.

### 3.2. Soil Microbial Alpha and Beta Diversity Reveal Community Disruption and Stability Loss Under Disease-Induced Dysbiosis

Microbial diversity analyses demonstrated that disease pressure significantly disrupted the structure and stability of soil microbial communities in *Morchella* cultivation systems. Regarding alpha diversity, Shannon, Simpson, and Inverse Simpson indices all indicated markedly higher diversity and evenness in healthy root-adhering soils (H_R), with Qianxi and Yongsheng values reaching approximately 4.5 and 4.3, respectively. In contrast, diseased groups (D and D_R) exhibited significantly lower diversity (QX_D ≈ 3.70, QX_D_R ≈ 3.75; YS_D ≈ 3.90, YS_D_R ≈ 3.85), suggesting that pathogen invasion led to community simplification and biodiversity loss (Figure 2a).

Beta diversity analysis further confirmed the destabilizing effect of disease on community structure. NMDS ordination based on Bray–Curtis dissimilarities revealed tightly clustered microbial communities in healthy H_R samples, indicating compositional consistency and ecological stability. In comparison, diseased soils (D and D_R) showed wider dispersion and greater heterogeneity, reflecting a shift toward unstable and disordered microbial assemblies. Additionally, unplanted control soils (CK) formed distinct clusters separate from cultivated soils, highlighting the influence of *Morchella* cultivation practices on shaping soil microbiomes (Figure 2b).

Collectively, these results suggest that disease-induced dysbiosis not only reduces microbial diversity but also undermines community resilience, leading to ecological instability and increased vulnerability to external stressors. Maintaining microbial homeostasis in the rhizosphere is therefore critical for sustaining soil health and crop productivity.

### 3.3. Taxonomic Composition of Bacterial and Fungal Communities Across Soil Types

At the phylum level, bacterial communities were predominantly composed of *Proteobacteria*, *Acidobacteria*, and *Gemmatimonadetes*, which collectively accounted for the majority of taxa across all samples (Figure 3a). Although *Proteobacteria* appeared more abundant in healthy rhizosphere and root-adhering soils (H and H_R), its overall variation was not statistically prominent. In contrast, diseased soils (D and D_R) showed an increased relative abundance of *Bacteroidetes* and *Planctomycetes*, which may indicate a shift in microbial functions from nutrient mutualism to enhanced organic matter degradation under pathogen stress. Moreover, *Verrucomicrobia* and *Candidatus rokubacteria* were uniquely enriched in control soils (CK), suggesting these oligotrophic taxa might be better adapted to nutrient-poor environments lacking rhizosphere effects.

Fungal communities at the phylum level (Figure 3b) also showed health-dependent compositional shifts. *Ascomycota* was dominant in all soils, especially in healthy samples, whereas diseased samples, particularly the Yongsheng diseased root-adhering soil (YS_D_R), showed increased abundance of *Basidiomycota*, *Chytridiomycota*, and *Blastocladiomycota*. This shift likely reflects an expansion of saprophytic and opportunistic fungi under diseased conditions.

At the genus level, beneficial or health-associated bacterial taxa such as *Sphingomonas*, *Bradyrhizobium*, and unclassified *Gemmatimonadetes* were enriched in healthy soils, possibly contributing to carbon cycling, nitrogen fixation, and overall microbial stability (Figure 3c). In contrast, *Pseudomonas*, *Polaromonas*, and unclassified *Acidobacteria* were more abundant in diseased soils, which may reflect microbial responses to environmental perturbations and increased organic matter input.

Fungal genus-level profiles (Figure 3d) revealed that *Morchella* dominated in healthy soils, particularly in root-adhering samples, suggesting it may represent the prevailing cultivated lineage. Diseased soils, however, exhibited marked increases in potential pathogenic genera such as *Fusarium*, *Penicillium*, *Aspergillus*, and *Trichosporon*, which may be involved in fruiting body decay and disease progression. Additionally, moderate enrichment of ectomycorrhizal fungi such as *Rhizophagus* in diseased soils may reflect an increased host reliance on mutualistic interactions under biotic stress.

Together, these taxonomic patterns indicate that healthy soils are likely to support functional microbial complementarity and ecological homeostasis, whereas diseased soils are characterized by disrupted microbial balance, pathogen dominance, and increased degradative capacity.

### 3.4. Region-Specific Taxonomic Biomarkers Identified by LEfSe and Differential Abundance Analyses

To identify microbial taxa associated with *Morchella* cultivation health and disease status, we performed LEfSe and differential abundance analyses across different soil types and regions. In the Yongsheng dataset, LEfSe identified several bacterial biomarkers with LDA scores greater than 3, clearly distinguishing health-associated genera such as *Bradyrhizobium* and *Sphingomonas* from disease-enriched taxa like *Pseudomonas* and *Dyadobacter* (Figure 4a). For fungal communities, *Morchella* and *Penicillium* were consistently enriched in healthy root-adhering soils, while saprotrophs and opportunistic pathogens—including *Fusarium* and *Trichosporon*—were significantly overrepresented in diseased soils (Figure 4b,c).

To evaluate whether these patterns were regionally consistent, a separate LEfSe analysis was conducted on the Qianxi dataset (Appendix A). Similar taxonomic patterns were observed: healthy microhabitats were enriched in *Sphingomonas*, *Janthinobacterium*, and *Morchella*, whereas diseased soils harbored greater abundances of *Beauveria*, *Botryobasidium*, and other stress-tolerant fungi.

Collectively, these findings demonstrate that distinct bacterial and fungal biomarkers are consistently associated with *Morchella* health status across geographically diverse cultivation systems, providing valuable indicators for soil health monitoring and disease risk assessment in morel farming.

### 3.5. Contrasting Assembly Dynamics of Bacterial and Fungal Communities Revealed by Neutral Model Fitting

Microbial community assembly processes were assessed using Sloan’s neutral community model, applied separately to bacterial and fungal communities across the two cultivation regions (Figure 5a–d). Bacterial communities exhibited a strong fit to the neutral model in both Yongsheng (R^2^ = 0.79, Nm = 319,905; Figure 5a) and Qianxi (R^2^ = 0.748, Nm = 408,794; Figure 5c), indicating that stochastic processes—such as dispersal limitation and ecological drift—are the primary forces governing bacterial community assembly in these soils.

In contrast, fungal communities displayed substantial deviations from neutral expectations, particularly in Yongsheng (R^2^ = 0.122, Nm = 118; Figure 5b) and to a lesser extent in Qianxi (R^2^ = 0.578, Nm = 598; Figure 5d). The relatively poor model fits, coupled with a higher proportion of taxa falling outside neutral confidence intervals, suggest that deterministic processes—including host filtering, microhabitat heterogeneity, and pathogen-mediated selection—play a dominant role in shaping fungal community structures.

These contrasting assembly mechanisms highlight fundamental ecological differences between bacteria and fungi in *Morchella* cultivation soils, with bacterial communities primarily shaped by neutral, stochastic processes, while fungal communities are more strongly influenced by deterministic environmental filtering and biotic interactions.

### 3.6. Network Topology Reveals Disease-Induced Destabilization of Fungal Communities

Fungal co-occurrence networks revealed notable structural reorganizations between healthy and diseased soils across both production regions (Figure 6a–d). In the Yongsheng site, the healthy root-associated network (YS_H + H_R) exhibited slightly increased complexity compared to the diseased network (YS_D + H_R), as reflected by a higher number of edges (816 vs. 790), modularity (0.501 vs. 0.492), and average clustering coefficient (0.751 vs. 0.736). In the Qianxi region, these structural differences were more prominent, with the number of edges rising from 755 in diseased soils to 931 in healthy soils, modularity increasing from 0.541 to 0.610, and average clustering coefficient from 0.660 to 0.800, indicating enhanced network cohesion under healthy conditions (Figure 6a,b).

Notably, the fungal genus *Morchella* consistently served as a central node in the healthy networks of both regions, but appeared to shift toward a peripheral position under disease conditions, accompanied by a decline in node degree and connectivity (Figure 6c,d). This topological repositioning suggests that disease stress may disrupt beneficial interactions between *Morchella* and other fungal taxa.

Meanwhile, diseased networks harbored increased connectivity among several non-core or opportunistic genera (e.g., *Mortierella*, *Fusarium*, *Aspergillus*), potentially reflecting pathogen-driven ecological filtering and competitive release (Figure 6b,d).

These network-level changes collectively imply a disease-induced deterioration of fungal ecological organization, characterized by reduced connectivity, weakened modularity, and diminished clustering. Such structural simplification may compromise the resilience and functionality of the fungal community, making it more susceptible to further perturbations.

### 3.7. Functional Differentiation of Microbial Communities Highlights Health-Dependent Metabolic and Elemental Cycling Potentials

Functional profiling based on CAZy, COG, and KEGG databases, along with elemental cycling gene analysis, revealed distinct metabolic differentiation among microbial communities across soil health statuses and microhabitats. Functional beta diversity showed significant separation between groups (ANOSIM R = 0.615, *p* = 0.001), indicating the influence of host health and rhizosphere compartmentalization (Appendix A). CAZy annotation identified higher abundances of glycoside hydrolases (GHs) and carbohydrate-binding modules (CBMs), particularly GT5, GH15, and GH23, in healthy rhizosphere and root-adhering soils (Appendix A). COG-based analysis showed enrichment of genes involved in carbohydrate metabolism, energy production, and amino acid metabolism in healthy soils, whereas diseased soils exhibited higher levels of defense mechanisms, cell motility, and signal transduction (Appendix A).

KEGG functional analysis revealed pronounced functional divergence between healthy and diseased root-adhering soils across both regions. In diseased soils, stress-responsive and proliferation-associated pathways were markedly enriched, including DNA replication, RNA polymerase, protein export, and homologous recombination, reflecting enhanced microbial proliferation and genomic plasticity under pathogenic pressure. Additionally, increased abundance of phosphonate and phosphinate metabolism, nicotinate and nicotinamide metabolism, and penicillin and cephalosporin biosynthesis in diseased soils suggests intensified antimicrobial resistance potential and disrupted nutrient metabolism (Appendix A).

In contrast, healthy soils exhibited significantly higher abundances of functional modules associated with carbon metabolism and ecosystem homeostasis, such as methane metabolism, nitrotoluene degradation, and cyanoamino acid metabolism, indicating enhanced carbon turnover and detoxification potential. Moreover, enrichment of alanine, aspartate and glutamate metabolism, insulin signaling pathway, and cell cycle–Caulobacter in healthy soils highlights stable nutrient cycling and tightly regulated microbial growth. These functional signatures collectively underscore a shift from ecologically balanced and functionally diverse communities in healthy soils to stress-tolerant, defense-primed communities in diseased soils.

### 3.8. Integrative Remodeling of Elemental Cycles Driven by Disease Pressure Functional Attenuation of Carbon Cycling in Diseased Soils

In healthy soils, carbon metabolism is predominantly sustained by oxidative pathways, with key functional genes such as *acs*, *fae*, *mxaF*, and *cooS* highly enriched. These genes support efficient organic carbon decomposition and energy production. In contrast, diseased soils exhibited a substantial decline in the abundance of carbon metabolism-related genes, particularly those involved in carbon oxidation and methylotrophy (Figure 7a,b, Figure 8a,b, Appendix A). This pattern indicates that disease stress weakens the carbon metabolic capacity of soil microbial communities, potentially leading to redox imbalance and diminished energy utilization efficiency.

Nitrogen cycling displayed a general attenuation in diseased soils (Figure 7c,d, Figure 8c,d, Appendix A). Healthy compartments were enriched in nitrate reduction genes, including *narG*, *napA*, and *narH*, promoting nitrate assimilation and nitrogen retention. In contrast, diseased soils exhibited a marked reduction in most nitrogen metabolism genes, except for *nirD* (nitrite reductase) and *norB* (nitric oxide reductase), which showed moderate increases. This suggests a functional shift toward partial denitrification under disease stress.

While the nitrate assimilation pathway was impaired, the enrichment of *nirD* and *norB* implies enhanced conversion of nitrite (NO_2_^−^) to nitric oxide (NO) and nitrous oxide (N_2_O), but without the completion to dinitrogen (N_2_) due to the absence of *nosZ* upregulation. This incomplete denitrification potentially exacerbates nitrogen loss via gaseous emissions and contributes to redox imbalance, further undermining soil fertility and ecosystem stability.

Phosphorus cycling genes exhibited a distinct stress-induced remodeling pattern in diseased soils. In the Yongsheng region, the diseased group showed significant upregulation of *ppnK*, *surE*, *phoA*, and organic phosphorus degradation genes *phnA*, *phnI*, and *phnW*, indicating a microbial shift toward reliance on organic phosphorus mineralization and stress-responsive phosphorus acquisition mechanisms. Conversely, higher abundances of *ugpB* and *relA* in the healthy group reflected more efficient inorganic phosphate transport and homeostasis regulation (Figure 7e,f and Appendix A). A similar pattern was observed in Qianxi, where diseased soils exhibited elevated expression of internal phosphorus mobilization genes (*PK*, *ndk*) and organic phosphorus degradation genes (*phnP*, *phnE*, *phnJ*), whereas the healthy group was enriched in *ugpA*, indicative of a robust inorganic phosphate uptake system (Figure 8e,f and Appendix A).

Sulfur metabolism revealed region-dependent functional shifts in response to disease stress. In the YS region, diseased soils exhibited higher abundances of sulfur oxidation genes (*soxZ*, *soxX*), whereas healthy soils were enriched in *asrA* and *sat*, genes associated with sulfite reduction. This may suggest that diseased communities partially maintain sulfur oxidation functions, while healthy communities favor reductive sulfur pathways (Figure 7g,h and Appendix A).

In the QX region, the diseased group was primarily enriched in *soxZ*, whereas healthy soils showed elevated abundances of *dsrA*, *dsrB*, *sqr*, and *phsA*, which are involved in dissimilatory sulfite reduction and sulfur redox buffering. This functional divergence may reflect different microbial strategies for sulfur transformation under varying stress conditions, though the extent to which these shifts directly influence soil health remains uncertain (Figure 8g,h and Appendix A).

Across both regions, the functional profiles of carbon, nitrogen, phosphorus, and sulfur cycling consistently showed that healthy soils maintain oxidative, energy-efficient metabolic states that support nutrient retention and ecological stability. In contrast, diseased soils exhibited functional reprogramming characterized by enhanced anaerobic metabolism, nitrogen loss through denitrification, phosphorus stress responses, and reductive sulfur metabolism. These patterns reflect systemic microbial dysbiosis and impaired elemental cycling under disease pressure, potentially undermining soil health and *Morchella* productivity.

## 4. Discussion

In recent years, although *Morchella* cultivation has expanded rapidly in China, persistent soil-borne diseases have emerged as a major constraint on yield stability and product quality. [37]. Documented diseases include cap dry rot caused by *Diploöspora longispora* [38,39], white mold by *Paecilomyces penicillatus* [39,40], stipe rot by *Fusarium* spp. [41], white rot by *Aspergillus* spp., and cobweb disease caused by *Cladobotryum protrusum* [42,43]. These pathogens often lead to malformed, decayed, or shriveled fruiting bodies, resulting in severe yield losses and economic damage. Their high adaptability, rapid proliferation under warm and humid conditions, and long-term persistence in soils make traditional control strategies largely ineffective [40]. This underscores the urgent need to unravel the microecological differences between healthy and diseased soils, particularly within rhizosphere and root-adhering compartments, to better understand pathogen invasion dynamics and develop sustainable disease management strategies.

### 4.1. Microbial Community Shifts Reveal Disease-Associated Disruption of Microbial Homeostasis

Our findings reveal that both geographic location and soil compartment (rhizosphere vs. root-adhering soil) significantly influence microbial community composition, with clear structural separation between healthy and diseased soils. Healthy root-adhering soils (H + R) exhibited higher alpha diversity and stronger network connectivity, suggesting that pathogen invasion disrupts not only community richness and evenness but also key cooperative interactions. This destabilization likely creates ecological “vacuum niches” that are readily exploited by opportunistic pathogens.

Consistent with prior research, we observed that disease stress suppressed beneficial taxa and weakened microbial network cohesion, leading to simplified, fragile community structures [17,19,44,45]. Such dysbiosis likely diminishes the soil’s ecological resistance and functional resilience against pathogen proliferation.

### 4.2. Deterministic Assembly and Network Destabilization Reveal Microbial Vulnerability Under Disease Pressure

The contrasting assembly mechanisms between bacterial and fungal communities underscore fundamental ecological differences in how these microbial groups respond to environmental pressures in *Morchella* cultivation soils. The strong fit of bacterial communities to the neutral model in both regions indicates that stochastic processes—including dispersal limitation and ecological drift—play a dominant role in shaping bacterial community composition. This pattern is widely observed in bulk soil ecosystems, where bacteria often exhibit high functional redundancy and broader ecological niches that buffer them against deterministic environmental filters [46].

In contrast, the fungal communities—particularly in Yongsheng—deviated significantly from neutral expectations, suggesting that deterministic processes such as host filtering, microhabitat specificity, and pathogen-driven selection exert stronger influences. This aligns with growing evidence that fungi, especially those involved in plant-fungal symbioses or antagonistic interactions, are highly sensitive to biotic and abiotic filters [47]. The pronounced deterministic assembly in diseased soils likely reflects a breakdown in competitive balance, where opportunistic taxa expand at the expense of beneficial symbionts.

The network analysis further corroborates this interpretation. Healthy fungal communities exhibited higher connectivity, modularity, and clustering coefficients—Features that typically reflect ecological resilience, functional redundancy, and stable nutrient cycling [48,49]. The central positioning of *Morchella* within these healthy networks suggests its role as a keystone taxon, potentially orchestrating cooperative interactions within the rhizosphere.

However, disease pressure induced a substantial collapse in network complexity. The displacement of *Morchella* toward peripheral nodes, coupled with the rise in opportunistic taxa as new hubs, indicates not only a compositional shift but also a functional destabilization of the fungal community. Similar patterns have been observed in plant-pathogen systems where network simplification correlates with reduced resistance to secondary infections and impaired ecosystem functions [50,51].

Collectively, these findings support the hypothesis that disease-driven disruption of deterministic fungal assembly and network topology leads to diminished ecological resilience, making the soil microbiome more susceptible to functional collapse. Future work integrating temporal dynamics, meta-transcriptomics, and synthetic community experiments will be essential to disentangle whether these changes are causal drivers of disease progression or secondary consequences of pathogen invasion.

### 4.3. Disrupted Elemental Cycling Reflects Microbial Dysbiosis and Ecosystem Instability

This study reveals that pathogen-induced stress in *Morchella* cultivation soils leads to systematic remodeling of elemental cycling processes, including carbon, nitrogen, phosphorus, and sulfur metabolism. This remodeling is characterized by the attenuation of oxidative, energy-efficient pathways typical of healthy soils, replaced by enhanced fermentation, partial denitrification, phosphorus stress responses, and reductive sulfur metabolism. These shifts are accompanied by reduced energy efficiency, redox imbalance, and increased reliance on alternative nutrient sources such as organic phosphorus and intracellular phosphorus reserves [47,52,53].

Of particular concern is the upregulation of *nirD* and *norB*, coupled with the absence of nosZ, in diseased soils, indicating a transition toward incomplete denitrification. This functional shift likely accelerates nitrogen loss through gaseous emissions (NO and N_2_O), intensifies soil redox stress, and compromises nitrogen retention. Consequently, it could directly impair soil fertility and *Morchella* productivity. Similar functional degradation patterns—Characterized by microbial dysfunction and nitrogen loss—have been reported in pathogen-infected rhizospheres of pepper [51] and soybean [54].

Phosphorus cycling exhibits a similarly significant restructuring. Healthy soils rely on high-affinity inorganic phosphate transport systems (*pstA*, *pstS*, *ugpA*) and phosphate homeostasis regulators (*relA*). In contrast, diseased soils show marked enrichment in genes involved in organic phosphorus mineralization (*phnA*, *phnP*) and phosphorus stress response (*phoA*, *phoB*). This transition suggests that microbial communities shift toward utilizing organic phosphorus and mobilizing intracellular reserves under conditions of inorganic phosphate scarcity or environmental stress [52,55].

Sulfur metabolism displays region-specific remodeling. In Yongsheng, diseased soils exhibit higher abundances of sulfur oxidation genes (*soxZ*, *soxX*), while healthy soils are enriched in sulfite reduction genes (*asrA*, *sat*), suggesting divergent redox strategies between health states. In Qianxi, healthy soils show higher abundances of *dsrA*, *dsrB*, *sqr*, and *phsA,* potentially indicating stronger capacities for sulfur reduction and redox buffering. These patterns imply that sulfur speciation and microbial sulfur transformations are influenced not only by disease pressure but also by local soil physicochemical properties [56,57].

Collectively, disease-induced microbial dysfunction affects not only community stability but also disrupts critical biogeochemical cycles and ecosystem service functions. Whether these functional shifts are primary drivers of disease development or secondary responses to pathogen invasion remains an open question. It is also unclear whether pathogen stress directly perturbs microbial metabolic networks, or whether the observed functional reprogramming represents microbial adaptation to environmental challenges such as redox imbalance and nutrient limitation [58,59].

Moving forward, targeted microbial restoration—such as introducing functional microbial consortia (e.g., sulfur-oxidizing bacteria or phosphate-solubilizing bacteria)—or modifying soil physicochemical properties (e.g., enhancing aeration, adjusting carbon-to-nitrogen ratios) may offer effective strategies to restore elemental cycling functions, thereby improving soil health and enhancing the productivity of *Morchella* cultivation systems [60]. Future studies should consider validating these findings with qPCR or enzyme activity assays.

## 5. Conclusions

This study demonstrates that soil-borne disease in *Morchella* cultivation disrupts microbial community structure, destabilizes functional networks, and rewires elemental cycling pathways, ultimately leading to ecological imbalance. These findings underscore the potential of microbiome-targeted interventions—such as functional microbial inoculants, habitat management, and soil restoration strategies—as promising approaches to enhance disease resilience and ensure sustainable productivity in *Morchella* farming systems.

Nonetheless, several limitations should be acknowledged. The present analyses were restricted to two geographic regions (Yongsheng and Qianxi), which, while representing distinct environmental and agricultural contexts, inevitably constrain the generalizability of our conclusions. The observed microbial patterns, community assembly processes, and shifts in biogeochemical cycling may also have been shaped by local soil properties, climate, or cultivation practices, and thus may not fully reflect the diversity of conditions across China or other global *Morchella* production zones. Furthermore, physicochemical parameters such as pH, organic matter, and nutrient availability were not measured at the time of sampling, limiting the integration of microbiological and edaphic data.

Future research should therefore extend to broader geographic scales, incorporate diverse environmental gradients, and include quantitative soil physicochemical analyses to validate and refine these findings. Integrating metagenomics with metatranscriptomics and metaproteomics will further enable a more comprehensive understanding of microbial functional dynamics, ideally complemented by targeted validation through qPCR assays, enzyme activity measurements, and culture-based approaches. Finally, because putative pathogens in this study were inferred from relative abundance data, their roles remain correlative; experimental verification through isolation and pathogenicity testing will be essential to establish causal relationships. Together, such efforts will facilitate the identification of core microbial indicators and functional traits underpinning soil health, ultimately supporting the design of regionally adapted, microbiome-based disease management strategies for sustainable fungal agriculture.

## Figures and Tables

**Figure 1 jof-11-00663-f001:**
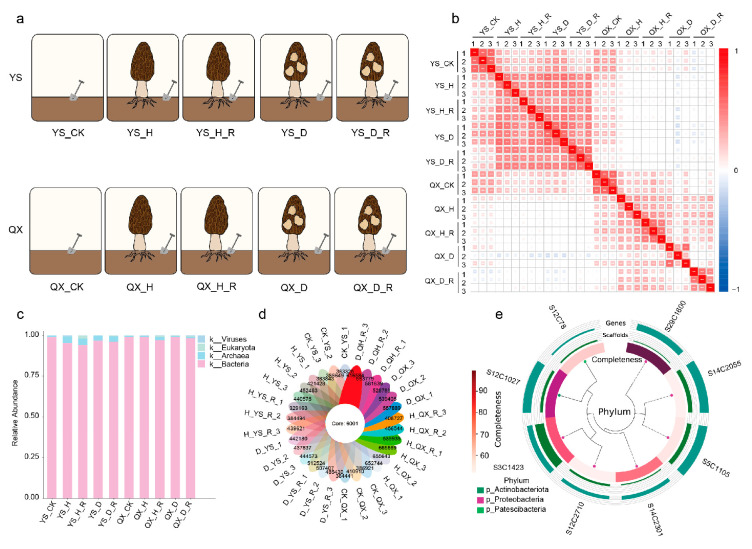
Experimental design and metagenomic dataset overview of *Morchella* cultivation soils. (**a**) Schematic representation of sampling types across two cultivation regions: Yongsheng (YS) and Qianxi (QX). Five distinct soil compartments were defined: CK—control soil from uncultivated adjacent fields; H—rhizosphere soil of healthy fruiting bodies; H_R—root-adhering soil from healthy fruiting bodies; D—rhizosphere soil of diseased fruiting bodies; D_R—root-adhering soil from diseased fruiting bodies. (**b**) Pairwise Spearman correlation heatmap of metagenomic samples based on non-redundant gene abundance profiles. (**c**) Taxonomic composition at the kingdom level based on Kraken2 annotations. (**d**) Venn circular plot showing the distribution of unique and shared non-redundant genes across all 30 samples. (**e**) Phylogenetic distribution and completeness of representative high-quality metagenome-assembled genomes (MAGs) recovered from the dataset.

**Figure 2 jof-11-00663-f002:**
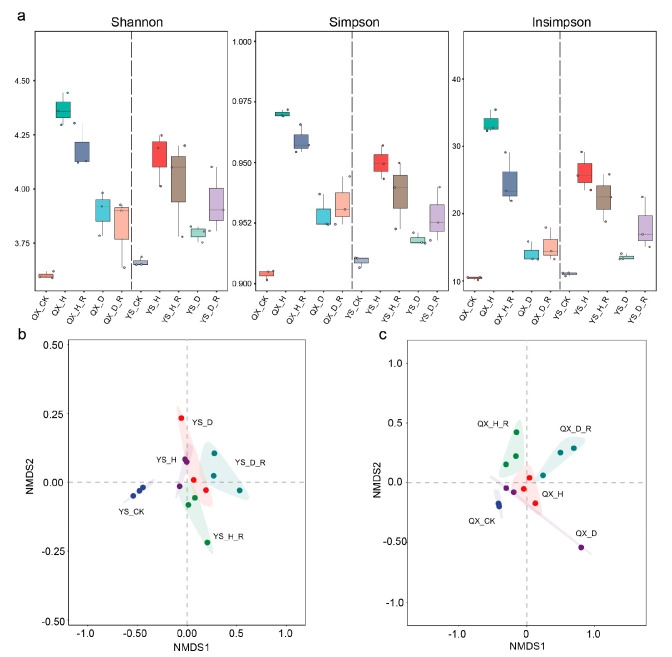
Microbial alpha and beta diversity patterns in *Morchella* cultivation soils under different health conditions. (**a**) Boxplots of three alpha diversity indices: Shannon, Simpson, and Inverse Simpson (Insimpson), calculated from non-redundant gene profiles. (**b**,**c**) Non-metric multidimensional scaling (NMDS) ordination based on Bray–Curtis dissimilarities of microbial communities. Ellipses indicate 95% confidence intervals for each group.

**Figure 3 jof-11-00663-f003:**
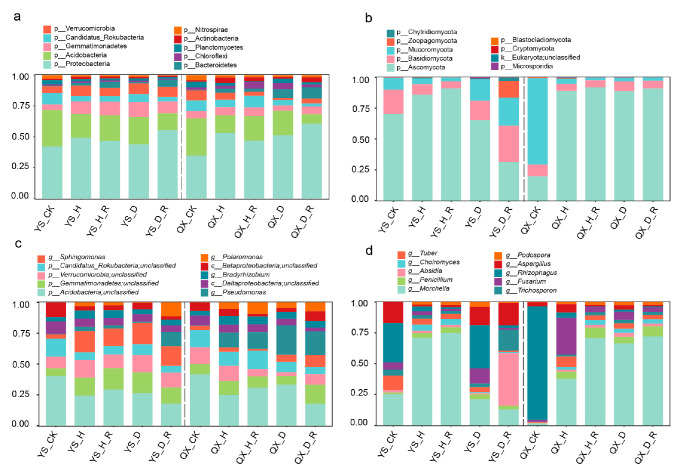
Taxonomic composition of bacterial and fungal communities across different soil compartments and health conditions in *Morchella* cultivation fields. (**a**,**b**) Relative abundance of dominant bacterial (**a**) and fungal (**b**) phyla in ten experimental groups. (**c**,**d**) Relative abundance of dominant bacterial (**c**) and fungal (**d**) genera.

**Figure 4 jof-11-00663-f004:**
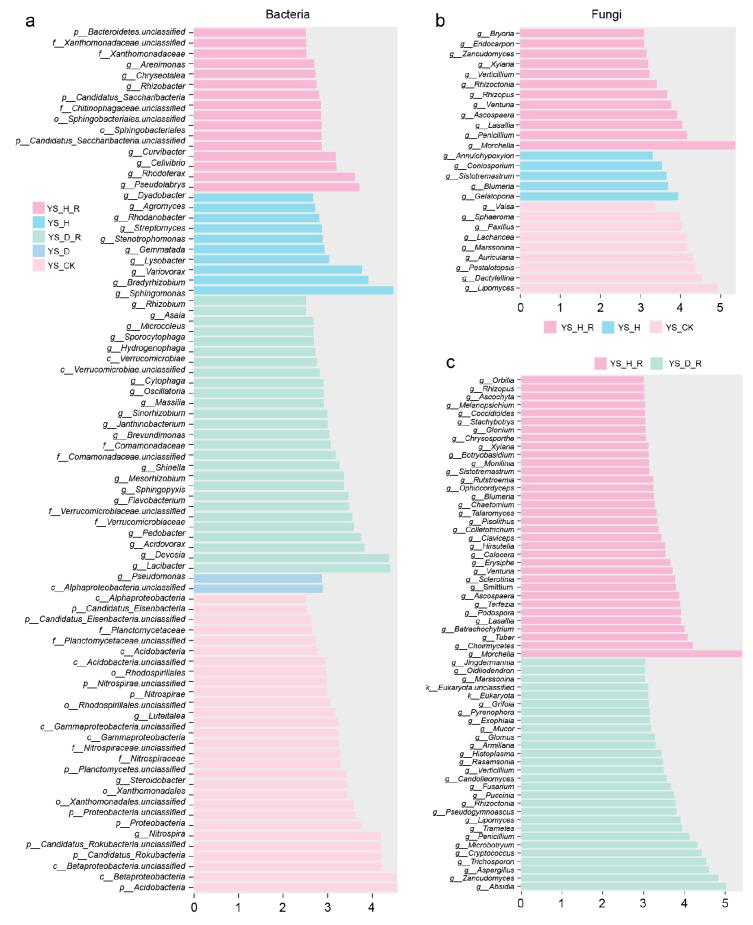
LEfSe-based identification of microbial biomarkers under different soil conditions in Yongsheng region. (**a**) Differential bacterial taxa identified across five treatment groups (YS_CK, YS_H, YS_H_R, YS_D, YS_D_R). (**b**) Differential fungal taxa between *Morchella* cultivation groups (YS_H, YS_H_R) and unplanted control (YS_CK). (**c**) Differential fungal taxa between diseased root-adhering soils (YS_D_R) and healthy root-adhering soils (YS_H_R).

**Figure 5 jof-11-00663-f005:**
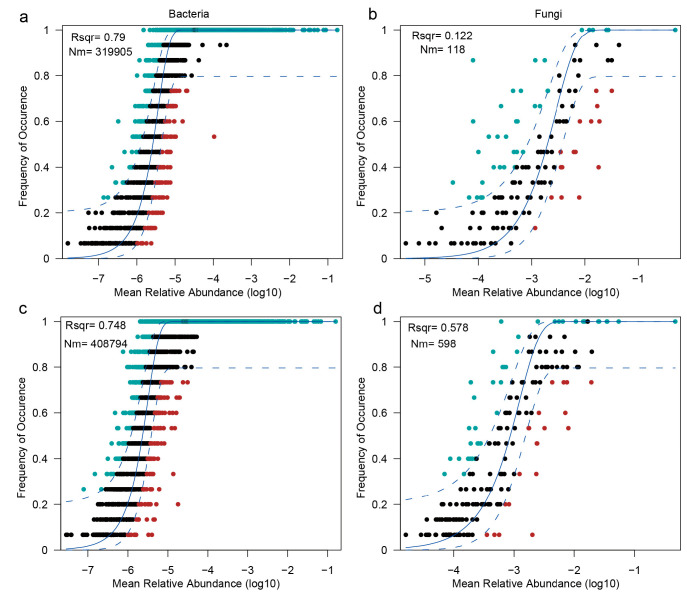
Sloan neutral community model fitting of bacterial and fungal communities. (**a**,**b**) Neutral model fits for Yongsheng (YS) region: (**a**) bacteria, (**b**) fungi. (**c**,**d**) Neutral model fits for Qianxi (QX) region: (**c**) bacteria, (**d**) fungi. Each panel shows the relationship between the log-transformed mean relative abundance (*x*-axis) and the frequency of occurrence across samples (*y*-axis). Black dots represent taxa that fall within the model prediction; blue and red dots denote overrepresented and underrepresented taxa, respectively. Solid blue curves represent model fit; dashed lines denote 95% confidence intervals. R^2^ indicates goodness of fit; Nm represents the estimated immigration rate.

**Figure 6 jof-11-00663-f006:**
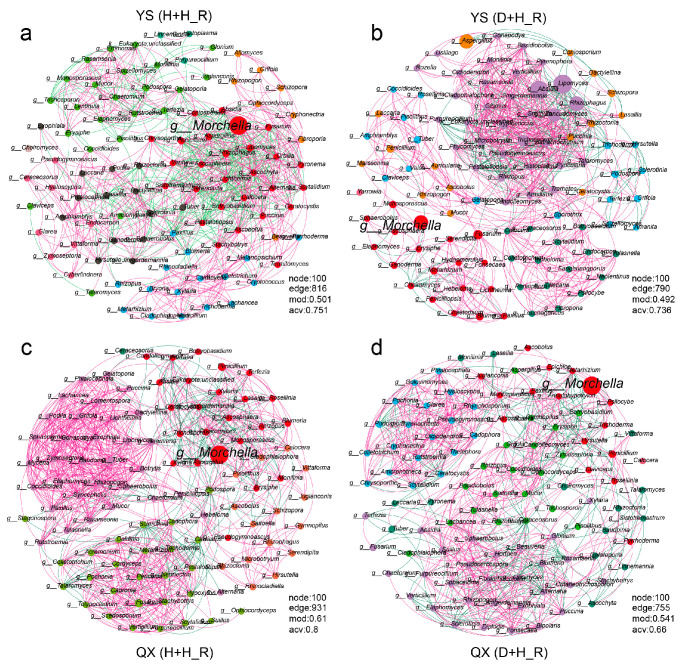
Fungal co-occurrence networks across different health conditions and regions. (**a**) YS_H_R (Yongsheng healthy root-adhering soil), (**b**) YS_D_R (Yongsheng diseased root-adhering soil), (**c**) QX_H_R (Qianxi healthy root-adhering soil), (**d**) QX_D_R (Qianxi diseased root-adhering soil). Each node represents a fungal genus, and edges indicate significant pairwise correlations (Spearman’s r > 0.5, *p* < 0.05). Node size is proportional to the relative abundance of each genus. Positive and negative correlations are shown in red and green, respectively. The genus *Morchella* is highlighted in red. Networks visualize changes in fungal interaction patterns under contrasting health conditions.

**Figure 7 jof-11-00663-f007:**
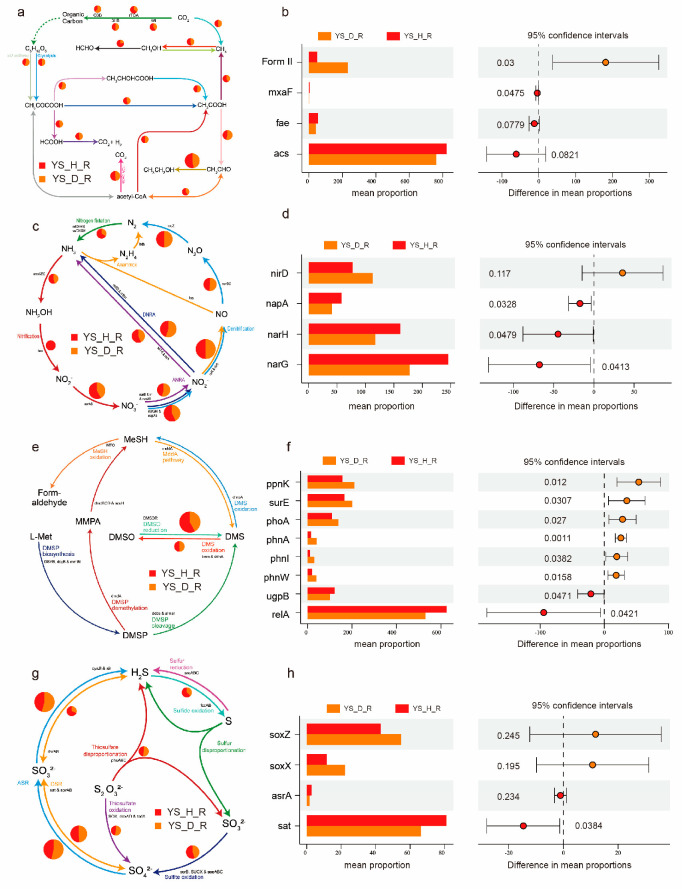
Disease-induced reprogramming of nutrient cycling pathways in the rhizosphere of *Morchella* spp. (Yongsheng region). (**a**,**c**,**e**,**g**) Schematic representations of microbial metabolic pathways related to carbon (**a**), nitrogen (**c**), phosphorus (**e**), and sulfur (**g**) cycling. Pathways were generated using Diting based on KEGG orthologs and gene annotations. Arrows indicate the direction of key biochemical transformations, and pie charts represent the proportional contributions of genes from healthy (YS_H_R) and diseased (YS_D_R) root-adhering soils. Key processes include carbon fixation, methane assimilation, denitrification, phosphate transport, and assimilatory/dissimilatory sulfur oxidation. (**b**,**d**,**f**,**h**) Differential abundance of representative functional genes between YS_H_R and YS_D_R soils, assessed by STAMP (Welch’s *t*-test, *p* < 0.05). (**b**) Carbon cycling: *Form II RuBisCO (Form II)*, *methanol dehydrogenase (mxaF)*, *formaldehyde-activating enzyme (fae)*, *acetyl-CoA synthetase (acs)*. (**d**) Nitrogen cycling: *nitrite reductase (nirD)*, *periplasmic nitrate reductase (napA)*, *nitrate reductase (narH, narG)*. (**f**) Phosphorus cycling: *polyphosphate kinase (ppk)*, *phosphatase (phoA)*, *alkaline phosphatase (phoN)*, *glycerol-3-phosphate permease (ugpB)*, *stringent response protein (relA)*. (**h**) Sulfur cycling: *sulfite oxidase (soxZ, soxX)*, *assimilatory sulfite reductase (asrA)*, *sulfate adenylyltransferase (sat)*. Right panels show 95% confidence intervals of the difference in gene proportions between groups.

**Figure 8 jof-11-00663-f008:**
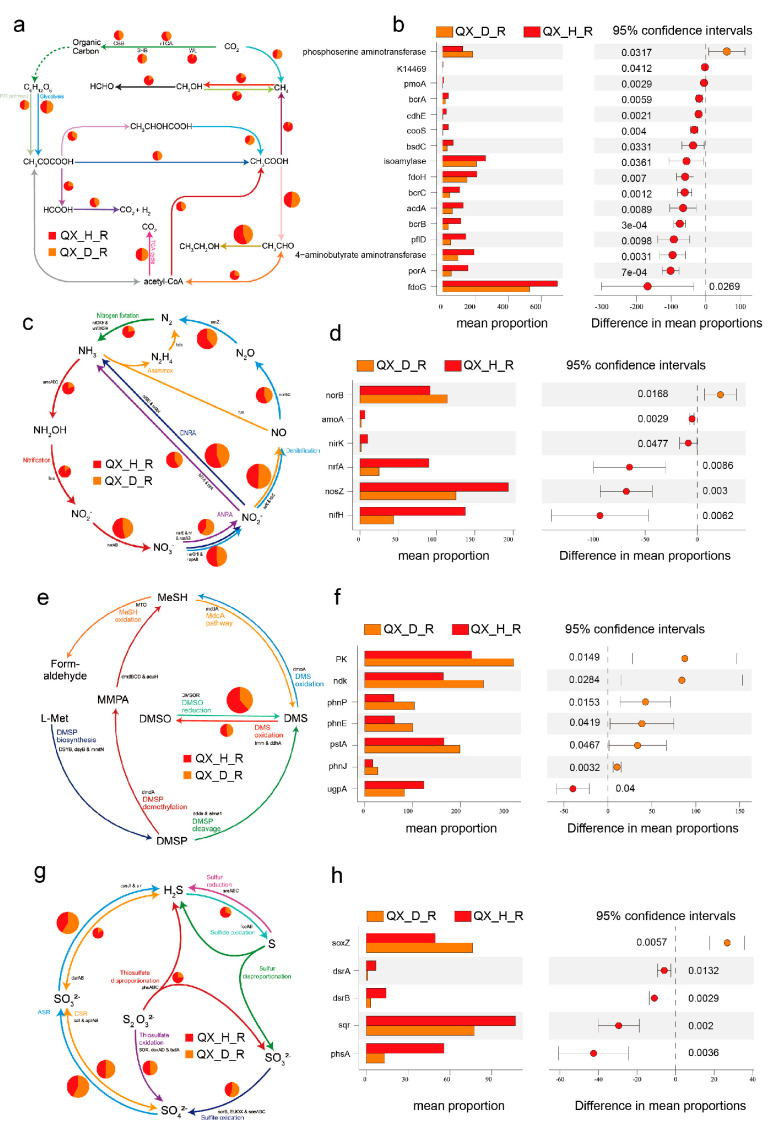
Disease-driven functional differentiation of nutrient cycling pathways in *Morchella* cultivation soils from the Qianxi (QX) region. (**a**,**c**,**e**,**g**) Schematic representations of microbial carbon (**a**), nitrogen (**c**), phosphorus (**e**), and sulfur (**g**) metabolic pathways based on KEGG ortholog annotations and visualized using Diting. Arrows indicate the direction of biochemical conversions, while pie charts display the relative gene contributions from healthy (QX_H_R) and diseased (QX_D_R) root-adhering soil samples. Major processes include glycolysis, carbon fixation, denitrification, polyphosphate transport, assimilatory sulfate reduction, and thiosulfate oxidation. (**b**,**d**,**f**,**h**) STAMP-based comparisons of key functional genes between QX_D_R and QX_H_R soils. (**b**) Carbon cycling genes: *phosphoserine aminotransferase*, *K14649*, *pmoA*, *brcA*, *cofH*, *ecpC*, *isoamylase*, *fdoG*, *acsA*, *ptfD*, *porA*. (**d**) Nitrogen cycling genes: *norB*, *amoA*, *nirK*, *nirF*, *nosZ*, *nifH*. (**f**) Phosphorus cycling genes: *polyphosphate kinase (ppk)*, *nucleoside-diphosphate kinase (ndk)*, *alkaline phosphatase (phoN)*, *phosphate transport genes (pstA, pstB, ugpA)*. (**h**) Sulfur cycling genes: *sulfite oxidase (soxZ)*, *dissimilatory sulfite reductase subunits (dsrA, dsrB)*, *adenylyl-sulfate reductase (aprA)*, *phosphoadenosine phosphosulfate reductase (phsA)*. Right panels present 95% confidence intervals of group-wise mean differences (Welch’s *t*-test, *p* < 0.05).

## Data Availability

The datasets generated and analyzed during this study are included in the article. Further inquiries can be directed to the corresponding authors.

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
