# Peer review of "Metagenomic Insights into Disease-Induced Microbial Dysbiosis and Elemental Cycling Alterations in Morchella Cultivation Soils: Evidence from Two Distinct Regions"

_jof, 2025, doi:10.3390/jof11090663_

Round 1

Reviewer 1 Report

This is an interesting and updated manuscript, exploring with advanced tools of molecular biology and microbial ecology (metagenomic) the disease-induced microbial dysbiosis and elemental cycling disruption in Morchella cultivation soils.  The authors elegantly connect basic and applied research with innovation, providing groundwork for the development of microbiome-informed strategies for a sustainable management of mushroom farming systems.

I really enjoyed reading this manuscript! I recommend its publication in the present form.

At the convenience of the authors, I would only suggest:

  1. In the final section of the introduction, when stating the objectives, avoid presenting them as separate sentences. Instead, list them sequentially, separated by semicolons, e.g. (i), (ii), etc.
  2. In the Discussion or Conclusions, maybe the authors would recommend for future studies, to validate key findings with qPCR or enzyme activity assays, as well as, to combine metagenomics with metatranscriptomics/ metaproteomics.
  3. In some figures (e.g. 1, 6,7) some letters/numbers are too small, but these are the output images generated by the bioinformatics tools used.

The main question addressed by this research is “How soil microbiomes in morel cultivation systems respond to health deterioration and pathogen invasion, particularly in terms of functional reprogramming, community stability, and nutrient cycling disruption?”

The methodology employed is internationally recognized for research of this nature, utilizing molecular biology and bioinformatics tools tailored to each research objective. The statistical analysis made in R is robust.

At the convenience of the authors, I would suggest to recommend in the Discussion or Conclusions, for future studies to validate key findings with qPCR or enzyme activity assays, as well as, to combine metagenomics with metatranscriptomics and metaproteomics.

The authors themselves acknowledge the limitation of the study related to that the research was based on samples from only two geographic regions. While these areas represent distinct environmental and agricultural conditions, the limited number of sampling sites may constrain the generalizability of their findings. The observed microbial patterns may not fully reflect the diversity of conditions across China or other global Morchella cultivation zones.

Author Response

Comment 1:
In the final section of the introduction, when stating the objectives, avoid presenting them as separate sentences. Instead, list them sequentially, separated by semicolons, e.g. (i), (ii), etc.

Response 1:
We thank the reviewer for this helpful suggestion. We have revised the final section of the Introduction to present the study objectives in a sequential format, using (i), (ii), (iii), etc., rather than listing them as separate sentences. This revision improves readability and conciseness. The updated text can be found on Page 2, Lines 81–90 of the revised manuscript:

“(i) Characterize changes in microbial taxonomic diversity and community composition in response to host health status and microhabitat variation; (ii) Identify region-specific microbial biomarkers and core functional modules that distinguish healthy from diseased soils, and elucidate the ecological processes—stochastic versus deterministic—that govern bacterial and fungal community assembly across different health states and microhabitats; (iii) Investigate how disease-induced microbial dysbiosis disrupts key nutrient cycling pathways, with a focus on carbon, nitrogen, phosphorus, and sulfur metabolism, thereby providing a foundational framework for microbiome-informed disease monitoring and sustainable management strategies.”

Comment 2:
In the Discussion or Conclusions, maybe the authors would recommend for future studies, to validate key findings with qPCR or enzyme activity assays, as well as, to combine metagenomics with metatranscriptomics/ metaproteomics.

Response 2:
We thank the reviewer for this excellent suggestion. We have revised the manuscript accordingly in both the Discussion and Conclusions sections to emphasize the importance of experimental validation and multi-omics integration. Specifically, in the Discussion (Lines 581–583), we added:“Future studies should consider validating these findings with qPCR or enzyme activity assays.”In addition, in the Conclusions (Lines 602–605), we further highlighted the future directions:“In addition, future research should integrate metagenomics with metatranscriptomics and metaproteomics to capture functional dynamics more comprehensively, alongside experimental validation through qPCR or enzyme activity assays.”

These additions address the reviewer’s comments by explicitly outlining methodological recommendations for future research, thereby strengthening the forward-looking perspective of the study.

Comment 3:
In some figures (e.g. 1, 6, 7) some letters/numbers are too small, but these are the output images generated by the bioinformatics tools used.

Response 3:
We greatly appreciate the reviewer’s comment. The relatively small font size in Figures 1, 6, and 7 is primarily due to the large number of microbial taxa presented, especially in the fungal co-occurrence network of Figure 6, which contains numerous nodes and edges. If all labels were enlarged, the overall layout would become overcrowded and less visually clear, thereby compromising the aesthetic quality of the figure and making the complex relationships more difficult to interpret. To maintain both clarity and overall visual balance, we chose to keep the current font scale so that readers can clearly grasp the structural patterns and core scientific information conveyed by the figures.

Closing Statement

Once again, we would like to express our sincere gratitude to the Editor and Reviewers for their insightful comments and thoughtful suggestions. We believe that the revisions made in response to these comments have significantly improved the clarity, scientific rigor, and overall quality of our manuscript. We hope that the revised version meets the journal’s standards and will be acceptable for publication.

Reviewer 2 Report

The manuscript addresses an important problem and employs a solid metagenomic approach, but interpretations overreach the data in several places, and methodological transparency could be improved.

Suggestions can be found as following:

Line 99: only two regions are studied, conclusion may not generalize to other Morchella cultivation systems. Please rephrase your statements

Line 103-106: There is a temporal limitation in sample collection, all of them were collected during fruiting stage, no time series data to assess progression of microbial changes. Please explain the reason or provide more data

Line 107: Please explain whether these triplicates are true biological replicates from independent plots or pseudo replicates within a single bed.

Line 109: There is no quantitative integration of soil physicochemical data (pH, organic matter, nutrient content) that could confound microbial patterns. Please add the information

Line 141: What is the identity/coverage thresholds beyond CD-HIT clustering?

Line 220: These pathogens are only inferred from relative abundance in metagenomic data. Better to do qPCR or culture-based confirmation.

Line 204: correct H_R to H+R

Line 614: What are these titles? Please indicate

All details can be found in the attached pdf file.

Author Response

Comment 1 (Line 99):
Only two regions are studied, conclusion may not generalize to other Morchella cultivation systems. Please rephrase your statements.

Response:
We agree with the reviewer that our conclusions may not be universally generalizable given the limited number of cultivation regions examined. We have revised the relevant statements in Discussion to emphasize that our findings represent case studies from two major production areas (Yongsheng and Qianxi). We also highlight that further cross-regional investigations will be required to validate and expand these patterns.

Comment 2 (Line 103–106):
There is a temporal limitation in sample collection, all of them were collected during fruiting stage, no time series data to assess progression of microbial changes. Please explain the reason or provide more data.

Response:
We appreciate the reviewer’s concern. The samples analyzed in this study were collected three years ago during the fruiting stage of Morchella cultivation. This stage was chosen because it represents the most critical period when soil-borne diseases become apparent and cause substantial economic losses. Unfortunately, due to the historical nature of these data, it is not possible to supplement time-series samples from earlier or later stages. We have now explicitly acknowledged this temporal limitation in Discussion sections, and we emphasize that future longitudinal studies covering multiple growth stages will be essential to capture the dynamic progression of microbial changes across the entire cultivation cycle.

In addition, it is important to note that this study was based on samples from only two geographic regions—Yongsheng and Qianxi. While these regions represent distinct environmental and agricultural conditions, the limited number of sampling sites may constrain the generalizability of our findings. The observed microbial patterns, assembly mechanisms, and shifts in elemental cycling could be influenced by local soil properties, climate, or cultivation practices, which may not fully represent the diversity of Morchella cultivation systems in China or worldwide. We have therefore revised the Discussion to highlight that further cross-regional studies encompassing broader geographic locations and environmental variables will be essential to validate and expand our conclusions. Expanding the spatial scale of sampling will also help identify core microbial indicators and functional traits associated with soil health, ultimately supporting the development of regionally adapted, microbiome-based disease management strategies for sustainable fungal agriculture.

Finally, we have added a statement to underscore that future research should integrate metagenomics with metatranscriptomics and metaproteomics to capture microbial functional dynamics more comprehensively and complement these analyses with experimental validation through qPCR or enzyme activity assays.

Comment 3 (Line 107):
Please explain whether these triplicates are true biological replicates from independent plots or pseudo replicates within a single bed.

Response:
We apologize for the lack of clarity. The triplicates represent true biological replicates, each collected from independent cultivation beds within the same farm region, separated by at least 5–10 meters to minimize microheterogeneity. We have clarified this point in the Materials and Methods section.

Comment 4 (Line 109):
There is no quantitative integration of soil physicochemical data (pH, organic matter, nutrient content) that could confound microbial patterns. Please add the information.

Response:
We sincerely appreciate the reviewer’s insightful comment. We acknowledge that soil physicochemical properties such as pH, organic matter, and nutrient contents are important ecological factors that can influence microbial community composition and functional patterns. However, in this study, such measurements were not collected at the time of sampling, and we recognize this as a limitation of our work. We have now explicitly acknowledged this limitation in the revised Discussion.

Despite this constraint, the primary focus of our study was to employ a metagenomic framework to characterize microbial community structure, functional potential, and elemental cycling pathways. This ecological dimension is often overlooked in prior research that has mainly emphasized soil physicochemical parameters. By adopting a metagenomic approach, our work provides novel insights into the microbial mechanisms underlying disease-associated dysbiosis in Morchella cultivation soils.

Moving forward, we agree that integrating metagenomic data with quantitative soil physicochemical measurements will be essential for a more holistic understanding of soil–microbiome–disease interactions. Such an integrative approach will enable the disentanglement of biotic and abiotic drivers of community shifts and help establish more robust ecological models. We have emphasized this point in the revised Discussion and suggested it as an important direction for future research.

Comment 5 (Line 141):
What is the identity/coverage thresholds beyond CD-HIT clustering?

Response:
Thank you for pointing this out. In the CD-HIT clustering step, open reading frames were clustered at 95% sequence identity and 90% alignment coverage to generate the non-redundant gene catalog. This information has now been explicitly added to the Materials and Methods.

Comment 6 (Line 220):
These pathogens are only inferred from relative abundance in metagenomic data. Better to do qPCR or culture-based confirmation.

Response:
We sincerely thank the reviewer for this insightful and valuable suggestion. We fully agree that inferring putative pathogenic taxa based solely on relative abundance in metagenomic data has inherent limitations, and that targeted validation through qPCR quantification or culture-based isolation would provide stronger, more robust evidence for their potential roles in disease dynamics.

Unfortunately, due to logistical and experimental constraints at the time of sampling—including the lack of immediate culture facilities and the absence of preserved subsamples under conditions suitable for downstream quantitative assays—our study was limited to inferences derived from metagenomic relative abundance data. As such, the associations reported here remain correlative rather than causative.

We have now explicitly acknowledged this as a key methodological limitation in the revised Discussion section. Furthermore, we emphasize that future studies should integrate metagenomic profiling with complementary approaches, including targeted qPCR, metatranscriptomics, and culturable microbiome analysis, to validate the functional activity and pathogenic potential of candidate taxa. Such a multi-omics and culture-integrated framework will be essential to advance from taxonomic correlations toward causal inference, thereby enabling a more mechanistic understanding of microbe–disease interactions in Morchella cultivation systems and supporting the development of microbiome-informed disease management strategies.

Comment 7 (Line 204):
Correct H_R to H+R.

Response:
We sincerely thank the reviewer for the careful observation. To avoid potential ambiguity and to ensure consistency, we have revised the group notations throughout the manuscript. The designations are now uniformly presented as: healthy rhizosphere (H), healthy root-adhering soil (H_R), diseased rhizosphere (D), diseased root-adhering soil (D_R), and unplanted control soils (CK). This adjustment has been implemented across the main text, figure legends, and supplementary materials for clarity and accuracy.

Comment 8 (Line 614):
What are these titles? Please indicate.

Response:
We sincerely thank the reviewer for the careful comment. In our submission, the supplementary figures and tables are provided as separate supplementary files, and their detailed content is included therein. To avoid redundancy, we used the general statement “Additional figures and tables are provided in the supplementary files.” This has now been clarified in the Supplementary Materials section

Closing Note

We once again thank the reviewer for these valuable comments and suggestions. All revisions have been incorporated into the manuscript, which we believe has significantly improved the overall rigor, clarity, and scientific contribution of our study.

Round 2

Reviewer 2 Report

The authors have revised the manuscript

according to all comments. 

I suggest to accept it.

The authors have made nice corrections. I suggest to accept it.